# Prediction of a Cyclic Hydrogenated Boron Molecule as a Promising Building Block for Borophane

**DOI:** 10.3390/molecules28031225

**Published:** 2023-01-26

**Authors:** Yasunobu Ando, Takeru Nakashima, Heming Yin, Ikuma Tateishi, Xiaoni Zhang, Yuki Tsujikawa, Masafumi Horio, Nguyen Thanh Cuong, Susumu Okada, Takahiro Kondo, Iwao Matsuda

**Affiliations:** 1CD-FMat, National Institute of Advanced Industrial Science and Technology (AIST), Tsukuba 305-8560, Ibaraki, Japan; 2Institute for Solid State Physics, The University of Tokyo, Kashiwa 277-8581, Chiba, Japan; 3RIKEN Center for Emergent Matter Science, Wako 351-0198, Saitama, Japan; 4Faculty of Pure and Applied Sciences, University of Tsukuba, Tsukuba 305-8573, Ibaraki, Japan

**Keywords:** boron, hydrogenated boron, calculation

## Abstract

We have extensively searched for a cyclic hydrogenated boron molecule that has a three-center two-electron bond at the center. Using first-principles calculations, we discovered a stable molecule of 2:4:6:8:-2H-1,5:1,5-μH-B_8_H_10_ and propose its existence. This molecule can be regarded as a building block for sheets of topological hydrogen boride (borophane), which was recently theoretically proposed and experimentally discovered. The electronic structure of the cyclic hydrogenated boron molecule is discussed in comparison with that of cyclic hydrogenated carbon molecules.

## 1. Introduction

The theoretical design of new molecules is a central issue in chemistry. Such methods have led to the syntheses of new functional materials such as catalysts or device components that have become technologically significant [1]. Molecules with unique bonding schemes have attracted academic interest because they often have unexpected chemical properties that violate well-established models and trigger innovations.

Boron is an electron-deficient element, and it has been found to form unique bonding configurations with multi-center atoms. Various boron-based molecules have been designed and synthesized [2]. Some of these molecules have shown anti-Van’t Hoff/Le Bel features [3,4,5,6]. The predicted existence of a 2D or planar boron material, borophene, has triggered vigorous explorations of its molecules or atomic sheets [7]. Researchers have reported growths of various borophene layers on metal substrates [8,9,10]. Theoretical and experimental searches for 2D boron have been extended to seeking low-dimensional materials composed of boron and other elements [7]. An example is the recent synthesis of sheets of hydrogen boride (HB) or borophane [11,12,13], which show intriguing functions such as hydrogen storage [14]. Boron and hydrogen atoms create a three-center two-electron (3c-2e) bond with unique chemical characteristics that allow us to design new 2D HB structures [15,16,17].

In this study, we searched for a possible planar structure of hydrogenated boron molecules (0D HB) through theoretical calculations. On the basis of the well-known boron hydride molecule diborone, shown in Figure 1a, we designed various molecular models by adding cyclic structures with different numbers of member atoms to different types of well-known boron-hydride (borane) molecules [18]. Among the twelve candidates, we found that only the molecule 2:4:6:8:-2H-1,5:1,5-μH-B_8_H_10_ was stable both thermodynamically and kinetically. The structural formula and atomic structure of the molecule are drawn in Figure 1b,c, respectively. The molecule has a 3c-2e bond at the center and two five-membered rings on the sides. The nomenclature of the molecule is based on references [19,20]; its atomic structure can be regarded as a building block of a topological borophane layer, discovered recently [12,13,15,17]. This research shows intriguing relationships between the planar molecules and atomic sheets of the hydrogenated boron compound. This cyclic hydrogenated boron molecule provides new perspectives in boron chemistry.

## 2. Results and Discussion

### 2.1. Candidates from the Cyclic Hydrogenated Boron Compounds

In searching for a new hydrogenated boron molecule, we examined the 12 candidates displayed in Figure 2 and listed in Table 1. We followed the same nomenclature as for the B_8_H_10_ molecule [19,20]. The atomic structures of the molecular candidates are schematically presented in Appendix A (Figure A1). The stable molecule was determined by filtering values of cohesive energy and signs of vibration frequency. All the candidates were found to gain cohesive energy during molecular formation. However, most of them were excluded because they had negative phonon energies or imaginary vibrational modes, indicating breakdowns in their molecular structure. It is apparent in Table 1 that only the molecule 2:4:6:8:-2H-1,5:1,5-μH-B_8_H_10_ (Figure 1b,c) is stable and could possibly exist in nature. The formation energy of this molecule is E_form_ = 194.5 kcal/mol, from four molecules of diborane—B_2_H_6_ (Table 2). Comparing this with E_form_ = 156.3 kcal/mol for eight molecules of borane (BH_3_) shows that the formation energy of the proposed molecule is 38.3 kcal/mol, which is slightly higher than that of the borane molecule.

A molecule of 2:4:6:8:-2H-1,5:1,5-μH-B_8_H_10_ has two types of boron atoms: those that make chemical bonds with hydrogen and those that do not. Two boron atoms in the 3c-2e bond at the center share hydrogen atoms. Besides the two boron atoms that connect with their neighbors, the other four boron atoms are individually terminated by two hydrogen atoms. To help understand the electronic states of the molecule, Figure 3 shows the schematics of the highest occupied and the lowest unoccupied molecular orbitals, HOMO and LUMO, respectively. Both orbitals extend inside the cyclic plane and have σ bonds.

### 2.2. Relation to the Borophane Layer

The molecule 2:4:6:8:-2H-1,5:1,5-μH-B_8_H_10_ has a five-membered boron ring on each of the two sides of the 3c-2e bond, as shown in Figure 1b,c. The topological borophane layer (α-HB sheet) reported recently also has such a pair of five-membered rings in its sheet structure, as shown in Figure 4a,b [12,13]. Among the different arrangements, the pair unit is the tiled β-HB sheet (Figure 4c,d). In this section, we discuss the physical and chemical relationships between the proposed molecule and borophane layers.

The α-HB sheet in Figure 4a,b was predicted by topological band theory and confirmed to be thermodynamically and kinetically stable by first-principles calculations [15,17]. It is noteworthy that its kinetic stability was supported by the positive vibrational energy of the phonon modes. This material was successfully synthesized through an ion-exchange reaction [12,13]. The α-HB sheet is a semimetal, with electronic states of the Dirac nodal loop (DNL) at the Fermi level. X-ray spectroscopic measurements have revealed a gapless electronic structure [12,13]. In Figure 4, the wave functions of (a) the lower and (b) the higher DNL bands at the Г point schematically overlap in the atomic structure model [15]. Focusing on the spatial distributions of the electronic states, one finds that the borophane layer can be described in terms of tiling with the molecular orbitals of the pair of five-membered rings. Wave functions of the energetically higher and lower DNL bands have out-of-plane (π) and in-plane (σ) characteristics, respectively. Instead of the molecular orbitals of 2:4:6:8:-2H-1,5:1,5-μH-B_8_H_10_, the shapes match the HOMO and LUMO of 2:3:4:6:7:8:-H-1,5:1,5-μH-B_8_H_8_, which is shown in Figure 4e,f.

A case of the β-HB sheet is given in Figure 4c,d; as the electronic structure and stability of the borophane layer have not been derived yet, we present our calculation results in Appendix B. The β sheet is also semimetallic, with a DNL at the Fermi level (Figure A2 in Appendix B). The phonon dispersion diagram for β-type borophane is shown in Figure A3. The phonon dispersion diagrams for β_1_-borophane show the imaginary phonon, while the one for β_2_-borophane shows that all the phonon energy is positive. The absence of an imaginary phonon modes implies kinematic stability, but it may not be sufficient to conclude their actual existence, as quadratic phonon bands appear near zero phonon energy. As shown in Appendix B, the electronic states of the β-HB sheet presented in Figure 4c,d are built with blocks of molecular orbitals of the pair of five-membered rings, as in the case of the α-sheet. The wave functions of the energetically higher and lower DNL bands have out-of-plane (π) and in-plane (σ) characteristics, respectively. The molecular orbital shapes are also similar to those of 2:3:4:6:7:8:-H-1,5:1,5-μH-B_8_H_8_ in Figure 4e,f.

Comparing Figure 3 and Figure 4 indicates that the electronic states of the borophane layer agree better with the HOMO and LUMO of 2:3:4:6:7:8:-H-1,5:1,5-μH-B_8_H_8_ than those of 2:4:6:8:-2H-1,5:1,5-μH-B_8_H_10_. The electronic properties can naturally be understood in terms of the bonding scheme between boron and hydrogen atoms. Besides the 3c-2e bond at the center, the rest of the B–H bonds in the B_8_H_8_ molecule are associated with one hydrogen atom, while those in B_8_H_10_ have two or no hydrogen atoms. The chemical environment of the B_8_H_8_ molecule generates the same π/σ LUMO/HOMO states as those of the borophane layer. It is worth mentioning that—in Table 1—2:3:4:6:7:8:-H-1,5:1,5-μH-B_8_H_8_ is unstable, although the HB sheets themselves are stable. We infer that the π electronic state is unfavorable for a single molecule of hydrogenated boron and that the molecule 2:4:6:8:-2H-1,5:1,5-μH-B_8_H_10_ consists of σ bonding states.

By further comparing wave functions between the B_8_H_8_ molecule and the DNL sheets of HB, one finds that the energy levels of the molecular orbitals are reversed between the π and σ types. It is intriguing to find this contrast although the molecule corresponds to a building block of the layer. To systematically investigate their electronic properties, this study calculated the energy difference E(π) − E(σ) for the molecules and the sheets. As shown in Figure 5a, the two types of low-dimensional structures—0D (molecule) and 2D (sheets)—have an opposite sign of energy difference E(π) − E(σ). This difference is apparently due to interactions between the building blocks in the 2D network. To examine the issue quantitatively, the tight-binding model was calculated using the hopping parameters between the composing boron atoms. An α-HB sheet structure was adopted in the simulation. The critical parameters were the transfer integrals—t_b_ and t_i_—at the two types of inequivalent 3c-2e bonds, where t_b_ and t_i_ correspond to the bonds at the unit center and those at the linkage between the units, respectively (Figure 5b). In the calculation, the transfer integrals at the single bonds between the boron atoms were uniformly set as t_s_. Under the condition t_b_/t_s_ = 1, the energy level of the π-orbital decreases by ΔE = −0.114 t_s_ when t_i_/t_s_ is changed only from 0 to 0.1. The electronic change in the molecular orbital through the t_i_ interaction is shown in Figure 5c. The isolated molecular orbitals (t_i_/t_s._ = 0) interact with each other and form a delocalized wave function (t_i_/t_s_ = 0.1) similar to the one in Figure 4a. The model calculation demonstrates that the energy level of the π molecular orbitals can become lower than that of a σ orbital through interactions between the building units. The development of the delocalized π states is likely responsible for forming the 2D network in the HB sheet.

The calculation results showed a significant role of the 3c-2e bond in the change in molecular orbitals between the cyclic molecule and topological borophane. Besides the electronic structure, it is interesting to consider a possible reaction path for building an HB sheet from HB molecules. Figure 6a recalls the molecular structure of 2:4:6:8:-2H-1,5:1,5-μH-B_8_H_10_. There are three types of boron atoms: B_C_, B_H_, and B_A_. The B_C_ atoms take part in the center 3c-2e B–H–B bonds and, thus, are not involved in linkages between molecules. The B_H_ atoms individually bond with two hydrogen atoms, while the B_A_ atoms are absent from the B–H bonding. Thus, we infer that a combination of B_H_ and B_A_ atoms naturally contributes to the two 3c-2e B–H–B bonds that connect the molecules that form the 2D network of the HB sheet.

To build an α-sheet composed of five-membered and seven-membered rings, one can assemble B_8_H_10_ molecules to form an intermediate seven-membered ring structure, as shown in Figure 6b. The heptagonal network is associated with two pairs of B_H_ and B_A_ atoms that naturally lead to the formation of 3c-2e bonds. There is also a pair of B_H_ atoms that can induce intermolecular B–H–B bonding by releasing two hydrogen atoms or H_2_. However, for a β-sheet, the intermediate seven-membered ring is formed by combinations of B_H_ atoms with combinations of B_A_ atoms, as illustrated in Figure 6c. Pairs of B_A_ atoms lack a hydrogen atom and are unlikely to generate a 3c-2e B–H–B bond themselves; thus, the B_8_H_10_ molecules cannot be used as blocks for building a β-structure. The argument for B–H–B bond formation favors α-sheet rather than β-sheet formation.

It is intriguing that β-borophane (Figure 4c,d) is not stable according to the calculations (Appendix B). However, the cyclic hydrogenated boron molecule 2:4:6:8:-2H-1,5:1,5-μH-B_8_H_10_ in Figure 1b,c and the topological α-borophane in Figure 1a,b were found to be stable. It is coincidental but intriguing to find that the reaction path of the molecular assembly favors a combination of the stable hydrogenated boron species.

Layers of topological borophane have been synthesized by liquid exfoliation associated with an ion-exchange reaction [11,12,13]. This is a top-down approach and typically results in powders of microscopic flakes. However, various 2D materials such as graphene and transition-metal dichalcogenide have been grown by chemical vapor deposition (CVD). This bottom-up approach has led to the synthesis of wide-area and single-crystalline films [21]. The cyclic B_8_H_10_ molecule proposed in this research has an atomic structure that corresponds to the building blocks of topological α-borophane. Thus, this molecule is the ideal precursor or CVD reacting gas for preparing single-crystalline topological borophane with a macroscopically wide area.

### 2.3. Orbitals of the Cyclic Boron Molecule in the Hückel Model

To examine the electronic states of a cyclic hydrogenated boron molecule, the classical Hückel model was applied and the results were compared with those for carbon molecules. The Hückel model has been used to simply calculate the energies of the π molecular orbitals in a conjugated carbon system by regarding all molecular orbitals as linear combinations of atomic orbitals [22,23,24,25,26,27]. Figure 7a shows the example of the cyclic conjugated carbon compound naphthalene (C_10_H_8_). This molecule consists of two loops of p_z_-orbitals, and each molecular orbital belongs to the genus-2 topological classification.

For comparison, we prepared a cyclic hydrogenated boron molecule with 10 boron atoms, corresponding to the 10 carbon atoms in naphthalene. The molecule is 1,6-1,6-decadiborane, B_10_H_10_, as illustrated in Figure 7b. As the two central boron atoms connect to each other through 3c-2e bonds, the p_z_-orbitals of the constituent boron atoms formed one molecular orbital loop, as shown in Figure 7b. This loop was topologically equivalent to a circle (genus-1), as illustrated in Figure 6c. This made it simple to calculate the Hückel model of the cyclic hydrogenated boron molecule. In this research, the Hückel model was calculated with a constant onsite Coulomb integral and constant resonance integral between neighboring atoms. The overlap integral was set to zero. A wave function calculated from the Hückel model is shown in Figure 7d for comparison with the LUMO in Figure 7e obtained from first-principles calculations. The molecular orbitals agreed well. Figure 7f,g also compares molecular orbitals between the Hückel model and the first-principles calculations for 2:4:6:8:-2H-1,5:1,5-μH-B_8_H_10_. Again, the molecular orbitals of the two models were similar. This similarity could be a theoretical test ground for deepening our understandings of quantum chemistry-based topological structures and electronic states.

## 3. Calculation Methods

Calculations of optimized atomic structures, molecular orbitals, and vibrational properties were all made via density functional theory (DFT), implemented using the Python package “Psi4 1.5.0” [28]. Becke three-parameter exchange and Lee–Yang–Parr correlations (B3LYP) were selected as an exchange-correlation functional [29,30]. The Gaussian basis set 6-311g(d,p) was applied to describe the electronic structures. Molecular geometry was optimized under Gaussian-level criteria.

For 2D HB or borophane sheets, calculations of wave functions and band diagrams were carried out via “Quantum ESPRESSO 7.1”, which is a first-principles code based on DFT, the plane-wave method, and the pseudopotential method [31]. The present calculations used the GGA–PBE exchange-correlation functional [32] with ultra-soft pseudopotentials in the library of standard solid-state pseudopotentials called “SSSP” [33]. The structure stabilities were evaluated from the phonon dispersion obtained from “Phonopy 2.16.3” [34], which can be used as post-process code for Quantum ESPRESSO.

The atomic structures and wave functions were visualized via Visualization of Electronic and STructural Analysis (VESTA) [35].

## 4. Conclusions

In summary, we predict the existence of a new cyclic hydrogenated boron molecule, 2:4:6:8:-2H-1,5:1,5-μH-B_8_H_10_, on the basis of first-principles calculations. This molecule can be regarded as a building block of topological hydrogen boride (borophane) sheets. This research shows intriguing relationships between the planar molecule and the atomic sheet forms of the hydrogenated boron compound. The cyclic B_8_H_10_ molecule provides new perspectives in boron chemistry. This molecule could also be the ideal precursor or CVD reaction gas for preparing wide-area topological borophane, which could lead to technological innovations.

## Figures and Tables

**Figure 1 molecules-28-01225-f001:**
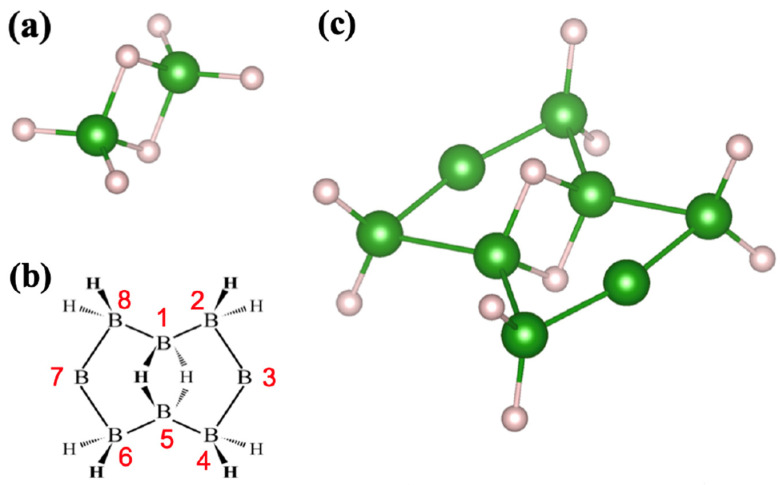
Molecules of hydrogenated boron compounds. (**a**) Atomic structure of diborane, B_2_H_6_. (**b**) A structural formula and (**c**) the optimized molecular structure of 2:4:6:8:-2H-1,5:1,5-μH-B_8_H_10_. Numbers of the boron atoms for nomenclature are labeled in (**b**).

**Figure 2 molecules-28-01225-f002:**
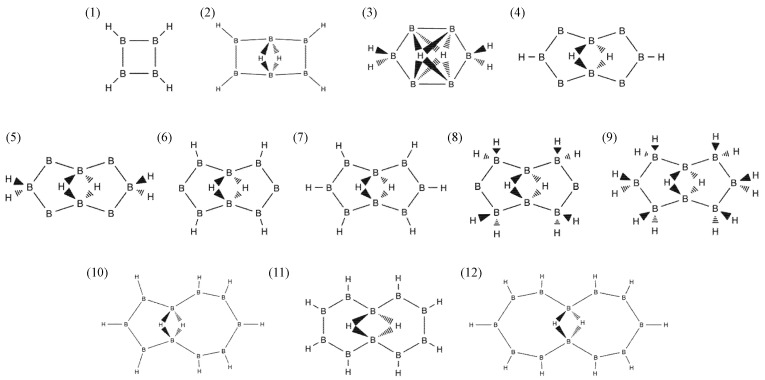
Structural formula of 12 candidates for a new cyclic hydrogenated boron molecule. A top-left number of each formula corresponds to the index in Table 1.

**Figure 3 molecules-28-01225-f003:**
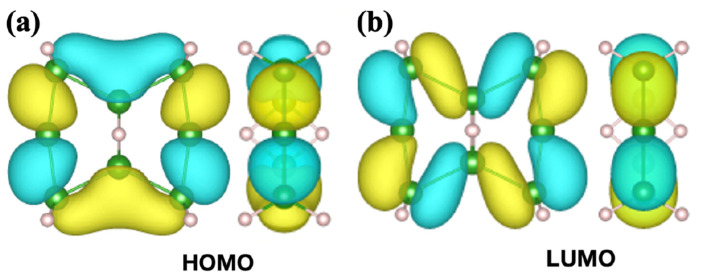
Top and side views of (**a**) the highest occupied molecular orbitals (HOMO) and (**b**) the lowest unoccupied molecular orbitals (LUMO) of 2:4:6:8:-2H-1,5:1,5-μH-B_8_H_10_. The HOMO and LUMO energy levels were −8.14 eV and −4.05 eV with reference to the vacuum level, respectively. The colors of the molecular orbitals correspond to the sign of their wave functions.

**Figure 4 molecules-28-01225-f004:**
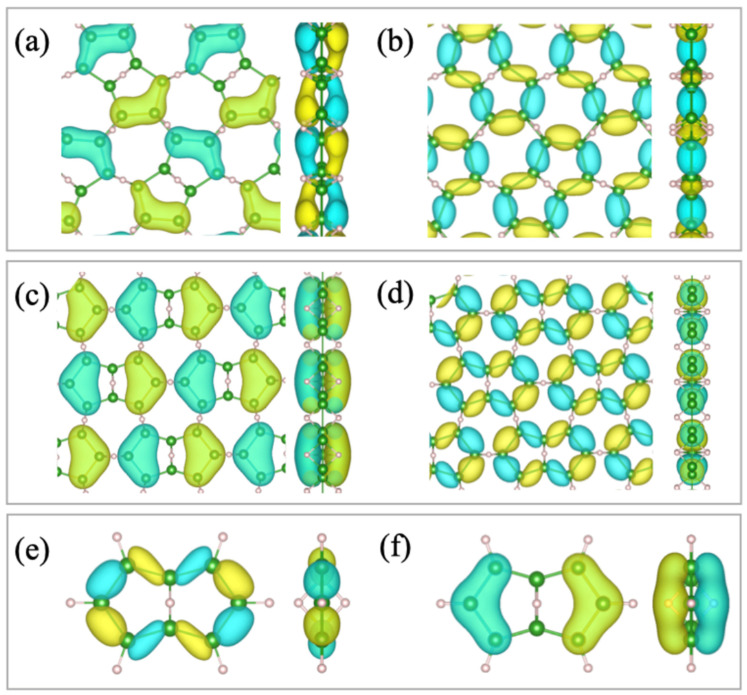
Atomic structures of the borophane layer (α_1_, β_1_), and the molecule (2:3:4:6:7:8:-H-1,5:1,5-μH-B_8_H_8_). Wave functions and molecular orbitals at Г point are schematically overlapped in the atomic models. (**a**) Lower and (**b**) higher bands of a Dirac nodal loop of the α_1_-borophane [15,17]. (**c**) Lower and (**d**) higher bands of a Dirac nodal loop of the β_1_-borophane (see Appendix A). (**e**) HOMO and (**f**) LUMO of the 2:3:4:6:7:8:-H-1,5:1,5-μH-B_8_H_8_ molecule. The color of each molecular orbital corresponds to the sign of its wave functions.

**Figure 5 molecules-28-01225-f005:**
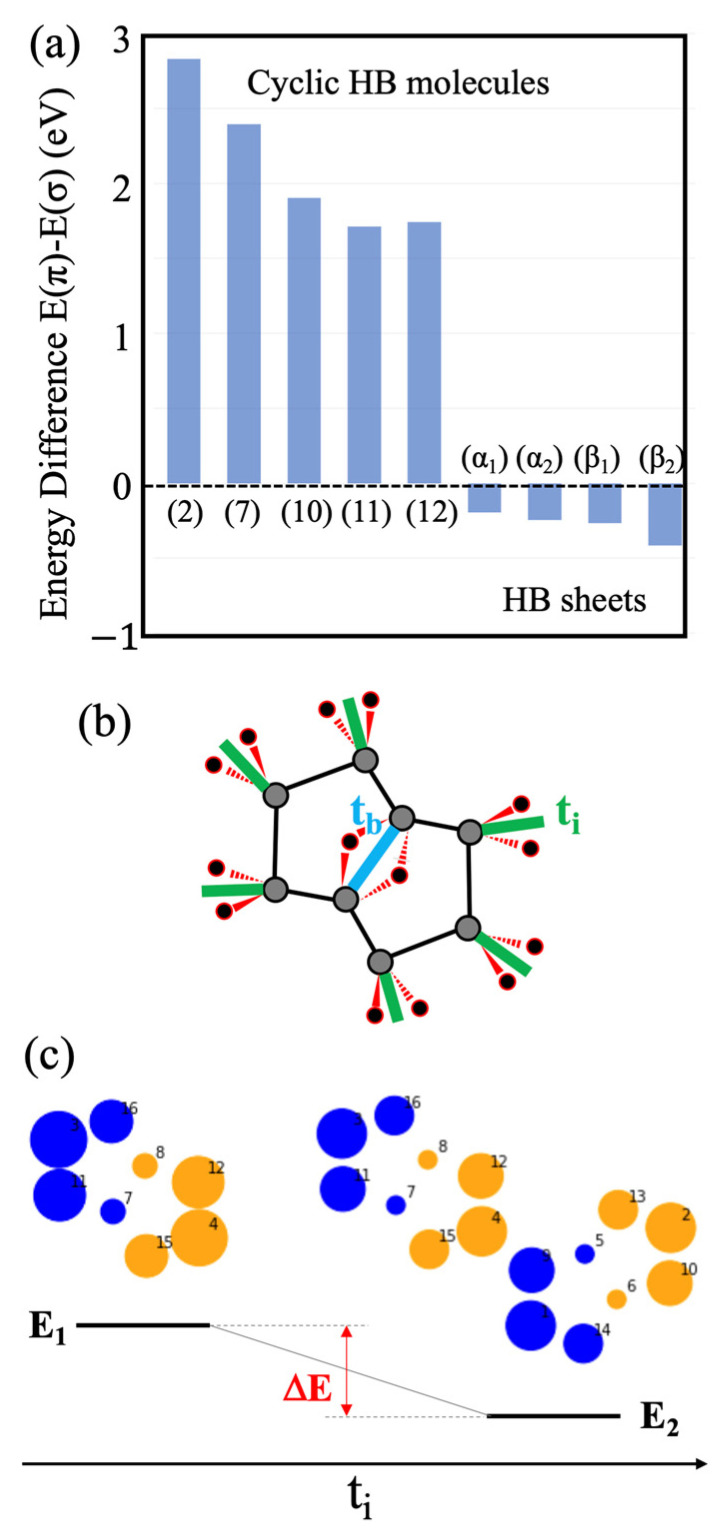
(**a**) Energy differences E(π) − E(σ) between boron materials, corresponding to the HOMO-LUMO gap (gap between lower-band and higher band of DNL at Г point) for the HB molecules (the HB sheets). Molecules: (2) 2:3:5:6:-H-1,4:1,4-μH-B_6_H_6_, (7) 2:3:4:6:7:8:-H-1,5:1,5-μH-B_8_H_8_, (10) 2:3:4:6:7:8:9:10:-H-1,5:1,5-μH-B_10_H_10_, (11) 2:3:4:5:7:8:9:10:-H-1,6:1,6-μH-B_10_H_10_, (12) 2:3:4:5:6:8:9:10:11:12:-H-1,7:1,7-μH-B_12_H_12_. The index numbers correspond to those in Figure 2 and Table 1. Sheets (at Г point): (α_1_) α_1_-borophane, (α_2_) α_2_-borophane, (β_1_) β_1_-borophane, (β_2_) β_2_-borophane. (**b**) Descriptions of the tight-binding parameters t_b_ and t_i_. (**c**) Energy diagram calculated from the tight-binding model with wave function forms before and after the interaction. Only one of the neighboring molecular orbitals is shown for the case t_i_/t_s_ = 0.1 for comparison with the results in Figure 4a.

**Figure 6 molecules-28-01225-f006:**
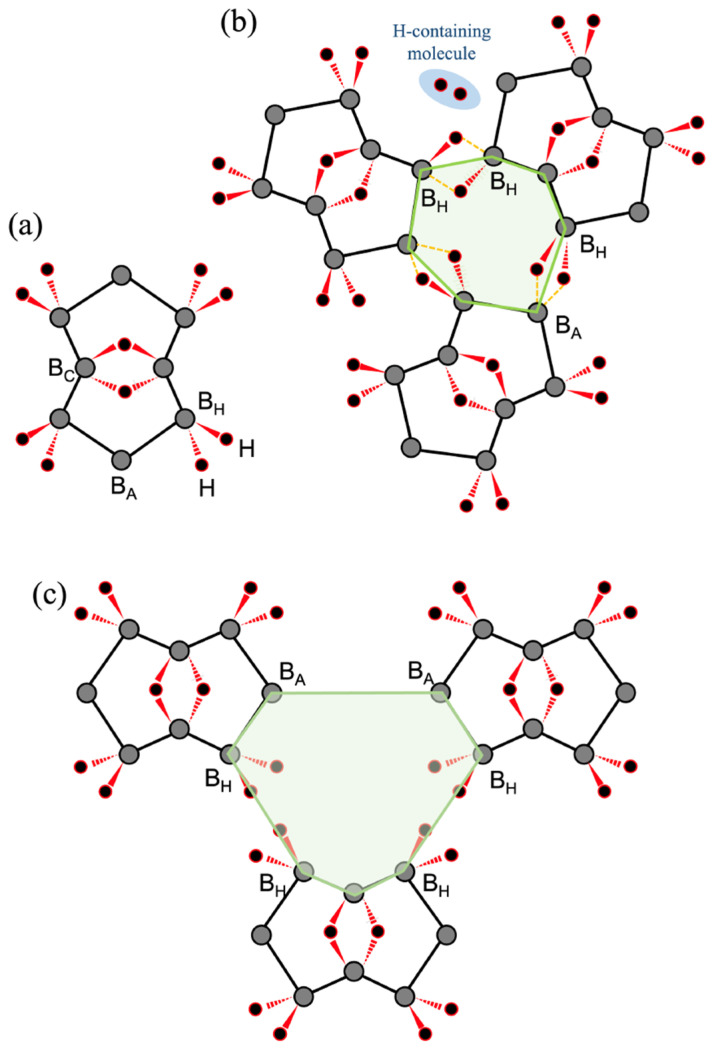
Construction of the borophane layer with the 2:4:6:8:-2H-1,5:1,5-μH-B_8_H_10_ molecules. (**a**) Molecular structures with labels on the different boron sites. (**b**) Building the α-borophane. (**c**) Building the β-borophane. Intermediate seven-membered rings are shaded in green in (**b**,**c**).

**Figure 7 molecules-28-01225-f007:**
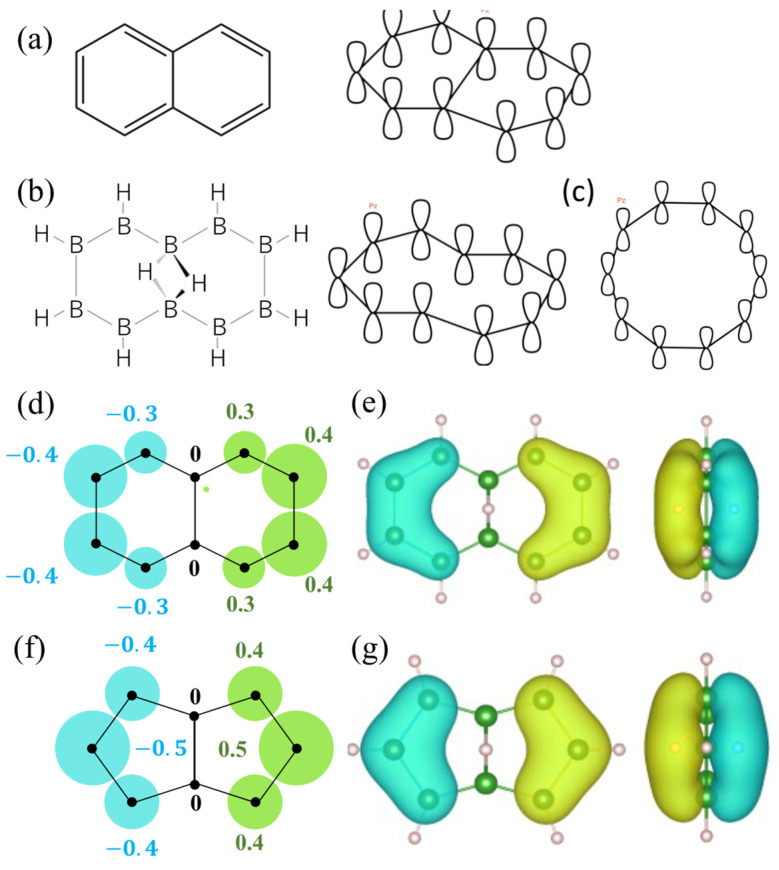
Application of the Hückel model to cyclic molecules. (**a**) Naphthalene molecule and the p_z_ configuration. (**b**) 1,6-1,6-decadiborane and the p_z_ configuration. (**c**) p_z_ configuration of a cyclic decagon molecule. (**d**,**e**) Molecular orbitals of 1,6-1,6-decadiborane, calculated from (**d**) the Hückel model and (**e**) first-principles calculations. (**f**,**g**) Molecular orbitals of 1,5-1,5-octadiborane, calculated from the (**g**) Hückel model and (**f**) first-principles calculations. The color of each molecular orbital corresponds to the sign of its wave functions.

**Table 1 molecules-28-01225-t001:** Filtration of the molecule candidates, TRUE or FALSE. Stability is determined by the absence (stability: TRUE) or presence (Stability: FALSE) of an imaginary vibrational mode. Atomic structures of the cyclic hydrogenated boron molecules are shown in Figure 2 and Appendix A.

Index	Molecule Candidate	Total Energy (eV)	HOMO Level (eV)	LUMO Level (eV)	Stability
1	1:2:3:4:-H-B_4_H_4_	−2768.32	−6.64	−2.50	FALSE
2	2:3:5:6:-H-1,4:1,4-μH-B_6_H_6_	−4148.76	−6.56	−3.73	FALSE
3	2:5:-2H-1,3,4,6:1,3,4,6-μH-B_6_H_6_	−4149.17	−7.65	−4.22	FALSE
4	3:7:-H-1,5:1,5-μH-B_8_H_4_	−5466.93	−5.93	−4.30	FALSE
5	3:7:-2H-1,5:1,5-μH-B_8_H_6_	−5499.07	−5.20	−4.54	FALSE
6	2:4:6:8:-H-1,5:1,5-μH-B_8_H_6_	−5501.27	−5.93	−4.16	FALSE
7	2:3:4:6:7:8:-H-1,5:1,5-μH-B_8_H_8_	−5533.90	−6.64	−4.24	FALSE
8	2:4:6:8:-2H-1,5:1,5-μH-B_8_H_10_	−5568.84	−8.14	−4.05	TRUE
9	2:3:4:6:7:8:-2H-1,5:1,5-μH-B_8_H_14_	−5635.07	−7.10	−3.70	FALSE
10	2:3:4:6:7:8:9:10:-H-1,5:1,5-μH-B_10_H_10_	−6915.86	−6.18	−4.24	FALSE
11	2:3:4:5:7:8:9:10:-H-1,6:1,6-μH-B_10_H_10_	−6917.17	−6.18	−4.46	FALSE
12	2:3:4:5:6:8:9:10:11:12:-H-1,7:1,7-μH-B_12_H_12_	−8299.69	−6.31	−4.57	FALSE

**Table 2 molecules-28-01225-t002:** B_2_H_6_, 8 BH_3_, and B_8_H_10_ + 7H_2_. All the systems have 8 boron and 24 hydrogen atoms.

System	Total Energy E (Hartree)	E − 4E(B_2_H_6_) (Hartree)	E − 4E(B_2_H_6_) (kcal/mol)
4 B_2_H_6_	−213.218	0.000	0.000
8 BH_3_	−212.969	0.249	156.250
B_8_H_10_ + 7H_2_	−212.908	0.310	194.528

## Data Availability

The data is contained in the article.

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
