# Peer review of "Prediction of a Cyclic Hydrogenated Boron Molecule as a Promising Building Block for Borophane"

_molecules, 2023, doi:10.3390/molecules28031225_

Round 1

Reviewer 1 Report

interesting work .

Author Response

We are pleased that you have interested in our work. We addressed the comments from other referees and revised our manuscript. The summary of change is also uploaded.

Thank you for your reviewing.

Reviewer 2 Report

This paper by Ando et al. is a fine and interesting work on predicting the cyclic hydrogenated boron molecule that can be a building block of borophane sheet. I would like to suggest the acceptance of this manuscript after a few minor revisions are made: 

1. The authors should expand the title. Since the large part of the paper is consideration of the predicted molecule as a building block for borophane, this should be reflected in the title.

2. There were 12 candidate molecules and only one proved to be stable. The structural formulas of all these molecules should be presented in the main text. The nomenclature in Table 1 is not really helpful in understanding their structure. 

Author Response

First, we appreciate your positive comments to improve our manuscript.

  1. The authors should expand the title. Since the large part of the paper is consideration of the predicted molecule as a building block for borophane, this should be reflected in the title.

***
It should be suitable to expand the title as the reviewer 2 commented above. We provided new title as follows:

“Prediction of the cyclic hydrogenated boron molecule promising as a building block for borophane”
***
2. There were 12 candidate molecules and only one proved to be stable. The structural formulas of all these molecules should be presented in the main text. The nomenclature in Table 1 is not really helpful in understanding their structure. 
***
We provided the structural formulas of 12 candidate molecules as Fig. 2. We shifted the figure numbers because of inserting this new figure. Additionally, we inserted a column of the index number on Table I to relate the Table I to other figures. All the revision is summarized in the document named “summary_of_change.pdf”.

We believe that our manuscript is now ready to publish on this journal.

Thanks again for your reviewing.

Reviewer 3 Report

To me, this paper are devoted to a quite interesting and popular trend, quantum chemistry of boron compounds. Results are obtained with present-days methods, and results sometimes look rather unexpected. Say, at the first glance I would never say that 2:4:6:8:-2H-1,5:1,5-161 μH-B8H10 will be more stable than 2:3:4:6:7:8:-H-1,5:1,5-μH-B8H8. Nevertheless I have no reasons to challenge these results. I believe it should be published as it is for wider discussion.  

Author Response

We are pleased that you gave us a quite positive recommendation. We addressed the comments from other referees and revised our manuscript. The summary of change is also uploaded.

Thank you for your reviewing.

Reviewer 4 Report

The proposed cyclic hydrogenated boron molecules for the building block of the topological borophane was interesting and I think the manuscript is deserved to be published in present form.

Author Response

(The authors gave the same response as above.)
